# Pathophysiology of COVID-19: A Post Hoc Analysis of the ICAT-COVID Clinical Trial of the Bradykinin Antagonist Icatibant

**DOI:** 10.3390/pathogens14060533

**Published:** 2025-05-27

**Authors:** Pierre Malchair, Jordi Giol, Javier Jacob, Jesús Villoria, Thiago Carnaval, Sebastián Videla

**Affiliations:** 1Emergency Department, Bellvitge University Hospital, University of Barcelona, Carrer de la Feixa Llarga, s/n, L’Hospitalet de Llobregat, 08907 Barcelona, Spain; jgiolamich@gmail.com (J.G.); jjacob@bellvitgehospital.cat (J.J.); 2Design and Biometrics Department, Medicxact, Plaza Ermita 4, Alpedrete, 28430 Madrid, Spain; carnavalthiago@medicxact.es; 3Neuropharmacology & Pain Group, Neuroscience Program, Bellvitge Institute for Biomedical Research (IDIBELL), L’Hospitalet de Llobregat, 08907 Barcelona, Spain; 4Clinical Research Support Area, Clinical Pharmacology Department, Germans Trias i Pujol University Hospital, Carretera de Canyet, s/n, Badalona, 08916 Barcelona, Spain; svidelaces@gmail.com; 5Fight Infectious Diseases Foundation, Badalona, 08916 Barcelona, Spain

**Keywords:** coronavirus infections, SARS-CoV-2, pandemics, bradykinin, inflammation

## Abstract

We used the data from a successful therapeutic assay that used icatibant in patients with hypoxemic COVID-19 pneumonia (the ICAT·COVID trial) to explore pathophysiological mechanisms. We performed concurrent-type, criterion-related validity analyses to assess the discriminative ability of a panel of nine potential serum markers (interleukin 6, ferritin, lactate dehydrogenase, C reactive protein, fibrin fragment D (D-dimer), complement 1 esterase inhibitor (antigenic and functional), complement 4 factor, and lymphocyte count) to predict the clinical milestones. Consistent with previous research, we evidenced a significant relationship between interleukin 6, lactate dehydrogenase and the lymphocyte count, and the clinical events. Furthermore, exposure to icatibant, a bradykinin B2 receptor antagonist (which improved pneumonia and mortality in the aforementioned randomised trial), attenuated this relationship, although this effect faded over time. The results reinforce the key role that the angiotensin-converting enzyme 2 has on COVID-19 pathophysiology as a point of convergence between the renin–angiotensin and kallikrein–kinin systems. This was shown clinically by the successful blocking of inflammatory pathways by icatibant at the bradykinin effector loop level early during the acute hyperinflammatory stage of the disease.

## 1. Introduction

In about three years, the severe acute respiratory syndrome coronavirus 2 (SARS-CoV-2) claimed approximately 7 million lives worldwide due to the so-called coronavirus disease 2019 (COVID-19) [1]. The waning of immune protection over time and the substantial number of immunosuppressed and other vulnerable individuals make it important to discover the pathophysiological pathways behind severe disease presentations.

The dysregulation of immune responses against SARS-CoV-2 is one of the main characteristics of COVID-19 pathophysiology [2]. Severe cases commonly feature lymphocytopenia, hyperinflammatory and prothrombotic states [3], and the development of a cytokine storm [4]. Although the SARS-CoV-2 infection triggers an innate immune response with common features, such as the activation of the pro-inflammatory interleukin 1 (IL-1) and IL-6 axis and complement-mediated opsonisation [2], experimental evidence shows that the renin–angiotensin (RAS), kallikrein–kinin (KKS), and coagulation systems are central in this derangement [5]. Moreover, a number of authors have suggested that the dysregulation of bradykinin (BK) could be key to understanding the acute respiratory distress syndrome seen in severe cases [6].

The tight regulation of vascular permeability is crucial to maintaining homeostasis. The interplay between RAS and KKS ensures this regulation and provides a compensating mechanism between inflammatory and tissue repair responses [7,8]. Both systems overlap at different levels; however, the role of the angiotensin-converting enzyme 2 (ACE2) is noteworthy. On the RAS side, it is part of the Mas Receptor pathway counter-regulatory branch, as it cleaves Ang-II into Ang-(1–7) (Figure 1), thereby exerting anti-inflammatory, anti-fibrotic, and vasodilating effects [9,10]. In addition, ACE2 cleaves Ang-I into Ang-(1–9), thus limiting the Ang-I availability to be converted into the pro-inflammatory peptide Ang-II (Figure 1) [11]. In turn, on the KKS side, ACE2 hydrolyses des-Arg-(9)-bradykinin (DABK), an active byproduct of bradykinin’s degradation [6,12], which is an agonist for the bradykinin B1 receptor (BKB1R) related to cytokine release, complement activation, vasodilation, and inflammation [10,13]. SARS-CoV-2 binding to the ACE2 for host cell entry downregulates its expression [14], impacting both systems additively or synergistically into a pro-inflammatory/oedematous state [8].

In a recent proof-of-concept study, we found that adding Icatibant (a competitive BKB2R antagonist) to standard care in patients with COVID-19 pneumonia admitted in the early hypoxemic stage was safe and improved both COVID-19 pneumonia and mortality [15]. Together with clinical milestones of interest, a panel of serum markers of inflammatory status was measured. Thus, it constituted a successful pharmacological assay from which pathophysiological lessons may be learned. The present manuscript concerns a post hoc analysis that evaluated the effects of this bradykinin blockade on the relationships between inflammatory markers and clinical milestones aimed at gaining insight into the pathophysiology behind severe cases of COVID-19 pneumonia.

## 2. Materials and Methods

### 2.1. Study Design, Patients, Procedures, and Outcomes

ICAT COVID (registered at ClinicalTrials.gov with the code NCT04978051) was a phase 2 proof-of-concept, randomised, open-label, controlled trial that assessed the safety and efficacy of icatibant to avert severe COVID-19 progression.

The study population consisted of 77 inpatients with COVID-19 pneumonia that required supplemental oxygen (the partial arterial oxygen pressure to the fraction of inspired oxygen ratio was below 380) but not high-flow oxygen or mechanical ventilation. They were recruited between 19 April 2021 and 8 February 2022, and randomly allocated to receive three 30 mg subcutaneous doses of icatibant per day for three consecutive days on top of standard care (icatibant group, *n* = 39) or standard care alone (SoC group, *n* = 38). The SoC included a variable combination of respiratory support measures, fluid therapy, antipyretic treatment, postural therapy, low-molecular-weight heparin, dexamethasone, remdesivir, and tocilizumab.

A complete panel of clinical outcomes was evaluated (see the Appendix A), including the time elapsed between the onset of symptoms and hospitalisation. We also gathered data on 10 dichotomic clinical milestones: (i) World Health Organization (WHO) score < 4 or ≥4 at visit 5 (V5); (ii) discharge ≤ V5 or >V5; (iii) partial (arterial) oxygen pressure (Pa_O2_)/inspiratory oxygen fraction (Fi_O2_) > 380 or ≤380 at V4; (iv) Pa/Fi > 380 or ≤380 at V5; (v) ROX index (the ratio between Pa/Fi and the respiratory rate) > 25 or ≤25 at V4; (vi) ROX index > 25 or ≤25 at V5; (vii) clinical response (see the Appendix A) at V5; (viii) clinical efficacy 28 days after the initial discharge; (ix) death due to COVID-19; (x) death by any cause. Moreover, we collected data on nine biomarkers that were used in this research: IL-6, ferritin, lactate dehydrogenase (LDH), C reactive protein, fibrin fragment D (D-dimer), complement (C) C1q (antigenic) inhibitor, C1q (functional) inhibitor, C4 factor, and lymphocyte count.

Further details on the design, participants, procedures, and outcomes can be found in the main publication of the ICAT·COVID trial [15].

### 2.2. Statistical Analysis

The levels of the nine biomarkers were described at each evaluation point and compared between the two subgroups defined by the ten dichotomic clinical milestones to explore their discriminative abilities. Generalised linear mixed models based on the Gamma distribution were used for this purpose. The individual trajectories of each marker were summarised using the best linear unbiased individual predictions to capture both the magnitude and the rate of changes into a single number (Appendix A). The best-suited markers to discriminate each milestone were objectively selected as those with variable influence on projection (VIP) values greater than 0.8 in the discriminant analysis based on the multivariate partial least squares regression of milestones over markers. Next, binary logistic regressions were used to ascertain the direction (either direct or inverse) of the associations between the milestones and markers. A formal analysis of the discriminative ability of the selected markers was then performed by calculating the area under the empirical receiver operating characteristic (ROC) curves to obtain pathophysiological clues. Lastly, the influence of the icatibant treatment over the discriminative ability was checked by modelling the marker distributions over the icatibant assignment both for the patients who attained and for those who did not attain each milestone and calculating the induced parametric binormal ROC curves for each of the conditions (see the Appendix A) [16,17]. Inferences on the ROC curves were performed using either nonparametric methods for empirical curves or asymptotic variance expressions in the binormal case. The last step was repeated for another covariate of interest, the time elapsed between the COVID-19 symptoms onset and hospital admission, which was deemed of pathophysiological interest. Since the latter was a continuous trait, singular induced ROC curves were calculated for two representative time points: 3 and 10 days.

## 3. Results

Of the 77 patients enrolled, 73 (37 and 36 of the active [icatibant plus SoC] and control [SoC alone] groups, respectively) completed all the study procedures and were used in this analysis. The patients in the active group were somewhat younger and more often male than the patients in the control group; otherwise, the two groups were well balanced with respect to baseline characteristics (Table 1). The respiratory physiology parameters were consonant with acute lung damage and impaired oxygen uptake. The clinical laboratory analyses showed mild lymphopenia, inflammation, and coagulation phenomena. Previous therapies, including vaccines, were also matched between the study groups. The clinical efficacy and deaths were significantly more favourable in the active group than in the control group (see the Appendix A).

IL-6, LDH, lymphocyte count, and C reactive protein were the markers that most diverged between the patients with favourable and unfavourable outcomes (Appendix A). Whilst IL-6, LDH, and lymphocyte count showed increasing divergences with time, C reactive protein showed the opposite pattern. From among all 90 (9 × 10) possible marker–milestone pairs, the discriminant analysis identified 38 as those best suited and worthy of further analyses (VIP > 8). In line with the previous step, the markers in these selected 38 pairs mostly concerned IL-6, LDH, C reactive protein, and lymphocyte count (Appendix A). With the exception of the lymphocytes, the association between the markers and clinical milestones was in general inverse for favourable outcomes and direct in the case of death (the higher the values, the lower the chance of clinical response or efficacy and the higher the chance of death). The opposite was true for lymphocytes (Figure 2).

In nearly all of these 38 pairs, the markers were found to have significant discriminative ability in the formal analyses (i.e., yielded areas under the ROC curves significantly greater than 0.5) (Figure 3 and Appendix A). Strikingly, the icatibant treatment attenuated the ability of the markers to discriminate the outcomes, which was denoted by significant changes in the areas under the ROC curves. There was an exception to this rule regarding the ability of IL-6 to predict the clinical efficacy 28 days after the initial discharge, which was enhanced rather than diminished (Figure 3, eighth row of first panel, in italics, and Appendix A). On the other hand, the time since the symptom onset enhanced the discriminative ability, with no marker–milestone pair exception. The results pertaining to the IL-6 are the most nuanced; the icatibant treatment somewhat deactivated the inverse association between IL-6 and the clinical milestones for the short-term outcomes (such as the Pa/Fi values at visit 4 or discharge by visit 5) but fostered the association between (lower) IL-6 levels and the long-term response. Nevertheless, this latter effect faded in the patients who took longer to go to the hospital. The interpretation of the results for other markers is more straightforward since there were no exceptions (the separation uniformly waned in the presence of icatibant and grew as the time since symptom onset increased).

## 4. Discussion

This post hoc analysis of a trial by our study group [15] found that IL-6, LDH, and lymphocyte count had outstanding discriminative abilities over different COVID-19 clinical milestones, which is consistent with findings from several other reviews [18,19]. The icatibant exposure had a somewhat nuanced but consistent effect on these relationships, providing pathophysiological clues and insight on how COVID-19 biomarkers can foretell prognosis.

Although this post hoc analysis was not driven by results, it was not pre-planned in the trial protocol, nor was the study specifically designed to elucidate pathophysiological mechanisms. Thus, although the parent trial was randomised, subsidiary causality between the putative biomarkers and clinical outcomes cannot be inferred, the statistical power may be inadequate, and the present report should be regarded as exploratory. Another limitation is that we could only assess a reduced panel of potential serum markers and we only had indirect statistical clues to unravel the pathophysiology. Alternative explanations for unrelated markers might include an excessive focus on pathways that are more amenable to being modified by icatibant. Nevertheless, it may be argued that events affecting alveolar gas exchange, such as the disruption of alveolar endothelium and lung oedema, may pose a more immediate life threat than other complications, whether thrombotic or otherwise. We did not assess the markers of endothelial function, such as soluble P-selectin or thrombomodulin, that together with D-dimer, were found to convey relevant prognostic information [20], nor did we measure the markers of immune status, which could have shed more light into the pathophysiology. We based our analyses on the alleged efficacy of icatibant, which others could not prove [21]. Although this might limit the external validity, the positive results of the parent trial [15] support the internal validity.

IL-6, a pleiotropic cytokine used to check the activation of acute inflammatory responses [22], showed an inverse relationship with the occurrence of favourable outcomes and a direct relationship with death. This is in consonance with the well-documented pro-inflammatory properties of this cytokine that have also been demonstrated in the COVID-19 context, in which it may act as a real-time indicator of an inflammatory response [10,23]. In turn, adjusting for icatibant attenuated its discriminative ability to predict short-term but not long-term clinical milestones, suggesting that the underlying cause-and-effect relationships have limited time spans. This notion is reinforced by the fact that the time from the symptom onset uniformly enhanced the associations between the IL-6 levels and clinical milestones, regardless of the presence of icatibant. Taken together, these results suggest that icatibant balanced the IL-6 effects by blocking the hyperinflammation elsewhere downstream, despite not antagonising IL-6 directly; however, this effect was lost as time went by, probably—and speculatively—because the lung tissue damage had gone beyond a point of no return, and because patients who continued to show high IL-6 levels in spite of therapy were those with the worst prognosis [24]. Intuitively, this no-return point would correspond to irreversible morphological changes or the induction of an immunosuppressed status [25]. Thus, blocking the hyperinflammatory state would make clinical sense before these changes begin.

Similar to IL-6, LDH, a biomarker of tissular necrosis [26], was inversely related to favourable outcomes and directly related to death. Again, adjusting for icatibant attenuated LDH’s discriminative ability to predict favourable outcomes, whilst adjusting for the time since the symptom onset strengthened it in the short term. Thus, we may reason that early exposure (≤10 days since symptom onset) to icatibant could revert hyperinflammatory damage. The findings for IL-6 and LDH are in line with a window of opportunity characterised by pre-apoptotic cell changes and membrane disruption, after which the downstream pharmacological blockade of the KKS would lose clinical effectiveness.

Unlike the two previous biomarkers, the lymphocyte count was directly related with the probability of favourable outcomes and inversely related with death, thus agreeing with previous literature findings [27]; notwithstanding, we must acknowledge that the white blood cell counts within our sample did not exceed typical levels of acute anti-infectious responses. Adjusting for both icatibant and the time since symptom onset either had no effect or somewhat attenuated the relationship between the lymphocyte count and clinical milestones, which was not surprising given that icatibant is not primarily an anti-infectious but rather an anti-inflammatory agent.

Some markers lacked discriminative ability, such as D-dimer, which was not particularly elevated and for which we were unable to find the correspondence with disease severity reported by a number of reviews [28]. However, the wider clinical context, in particular the comorbidities and thrombotic complications, might explain most of the correspondence between the D-dimer levels and COVID-19 outcomes [28]. Of note, increased D-dimer levels were typically seen in patients who developed coagulopathy, but not necessarily in those who did not [29,30]. This stresses the influence of factors other than altered fibrin turnover in the pathogenesis of acute respiratory distress. Similar reasoning can be used for ferritin [31]. In addition, C reactive protein has also been noted to be of prognostic relevance [32], but it precedes other pathophysiological events, such as complement activation or the production of pro-inflammatory cytokines [19]. Thus, the correspondence with the outcomes is less immediate [32]. Speculatively, these latter, more immediate events would have a greater ability to discriminate the clinical course in the absence of coagulopathy than acute-phase reactants, which are a shared feature of patients requiring hospitalisation as a whole [30]. In this vein, the C1q inhibitor did not emerge in our analyses—which should be no surprise, given that it was invariably elevated in all the patients regardless of the clinical course, the time since symptom onset, or allocation of icatibant (see Appendix A). Probably, C1q inhibitor elevation is an early event given its close relationship with SARS-CoV-2-mediated ACE2 downregulation and may even precede DABK accumulation and the triggering of BKB1R-related inflammatory cascades [2,8,10,13,33].

Several implications arise from this research. First, in line with what was reported in the introduction, bradykinin appears to be a pivotal point of convergence between RAS and KKS for COVID-19 pathophysiology, in turn supporting the pathophysiological role of ACE2 [34]. The effectiveness of the downstream blockade at the BKB2R level (i.e., close to the final effector steps of vasodilation and oedema production) speaks of a funnel effect for which more upstream interventions would be comparatively less efficacious and sheds light on the paradox that general anti-inflammatory therapies (i.e., corticosteroids) have proved more beneficial than selective kinase inhibitors and IL-6 inhibitors against COVID-19 [35]. Second, such a downstream blockade of these two systems would, in turn, lead to beneficial effects on other nonspecific inflammatory pathways, including the coagulation and contact systems (Figure 4), although these appeared to be less related to the clinical phenomena. This would also explain the success of icatibant in the parent trial [15]. Third, the apparent time sequence and limited span of these effects yield credibility because they are consistent with acute-phase phenomena that could only be reverted before permanent anatomical damage is established. Fourth, in line with previous contributions [18,19], IL-6, LDH, and lymphocyte count can be consolidated as prognostic biomarkers of the COVID-19 clinical course.

In conclusion, the pulmonary and systemic disruption of ACE2 seems to be the key to the harmful potential of SARS-CoV-2, and targeting bradykinin as a point of convergence between RAS and KKS may be more beneficial than more upstream interventions. The timing of the intervention (i.e., early during the hyperinflammatory stage, before the acute respiratory distress syndrome fully unfolds) also seems to be of relevance.

## Figures and Tables

**Figure 1 pathogens-14-00533-f001:**
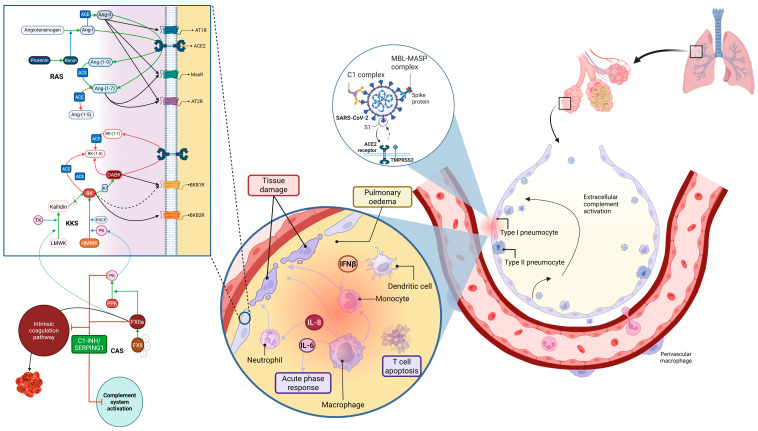
Summary of COVID-19 pathophysiology: postulates and data known before this study. SARS-CoV-2 triggers a local alveolar inflammatory response, with complement system activation, chemokine and cytokine releases, and white blood cell recruitment and degranulation. This growing inflammatory response causes tissue damage and starts a vicious positive feedback loop. A similar response occurs when the virus reaches the capillary bloodstream, damaging the endothelial cells and causing capillary leakage and pulmonary oedema. On the other hand, SARS-CoV-2 binding to the ACE2 receptor leads to its downregulation, which (i) decreases the Ang-(1–7) levels, and thus, reduces the ACE2/MasR pathway anti-inflammatory signalling, and (ii) also lowers the degradation of DABK (an active byproduct of BK), increasing the inflammatory signalling through the BKB2R. Additionally, several SARS-CoV-2 proteins interact with the C1 inhibitor, a main CAS regulator, reducing its inhibitory ability over the complement and kinin–kallikrein systems, and the coagulation system. Altogether, these snowballing inputs enhance the levels of circulating proinflammatory cytokines, causing the widely reported cytokine storm. Abbreviations: A, Aminopeptidase; ACE, Angiotensin-Converting Enzyme; Ang, Angiotensin; AT1R, Angiotensin II Type 1 Receptor; AT2R, Angiotensin II Type 2 Receptor; BK, Bradykinin; BKB1R, Bradykinin B1 Receptor; BKB2R, Bradykinin B2 Receptor; C1, Complement Factor 1; C1-INH/SERPING, C1 Inhibitor (encoded by the Serine Protease Inhibitor [SERPIN] family G member I); CAS, contact activation system; DABK, des-Arg^9^-Bradykinin; FXII, Coagulation Factor XII; FXIIa, Activated Coagulation Factor XII; HMWK, High-Molecular-Weight Kininogen; IFNβ, Beta Interferon; IL-6, Interleukin 6; IL-8, Interleukin 8; KI, Kininase I; KKS, Kinin–Kallikrein system; LMWK, Low-Molecular-Weight Kininogen; MASP, Mannan-Binding Lectin-Associated Serine Protease; MasR, Mas Receptor; MBL, Mannose-Binding Lectin; PK, Plasma Kallikrein; PPK, Pre-kallikrein; PRCP, Prolylcarboxypeptidase; RAS, Renin-Angiotensin System; SARS-CoV-2, Severe Acute Respiratory Syndrome Coronavirus 2; TK, Tissue Kallikrein; TMPRSS2, Transmembrane Serine Protease 2. Adapted from “Lung and Alveoli with Callout (Layout)” and “Alveolar-Capillary Barrier”, by BioRender.com (2023). Retrieved from https://app.biorender.com/biorender-templates (accessed on 30 August 2023).

**Figure 2 pathogens-14-00533-f002:**
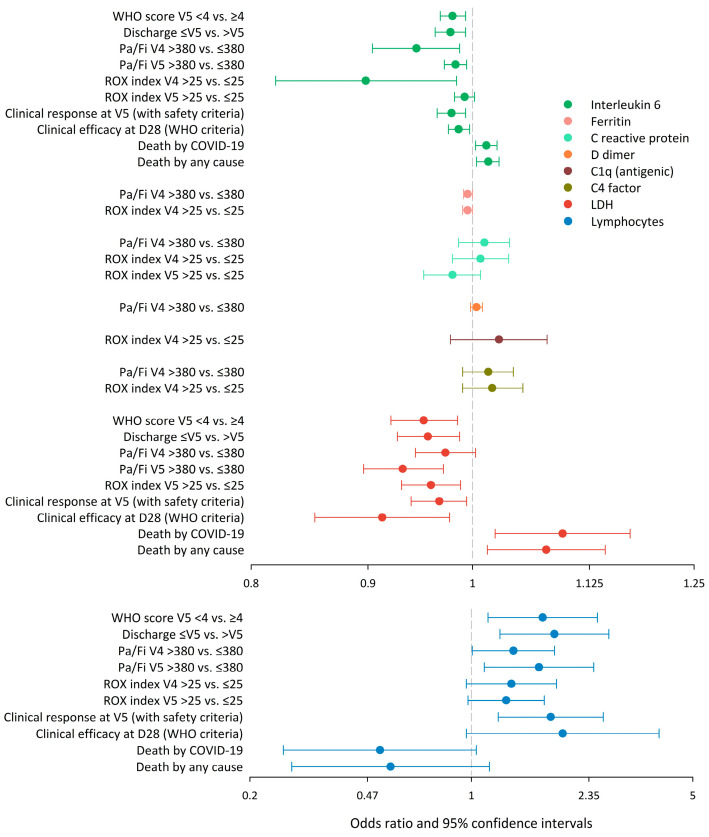
Summary of the association analyses (binary logistic regression) of milestones over markers (38 selected pairs) to learn the direction of the associations. The association was in general inverse for favourable outcomes and direct for the case of death. Thus, the higher the values of inflammatory markers, the lower the chance of a clinical response and the higher the chance of death. The opposite was true in the case of lymphocytes. Overlapping of some confidence intervals with the no-association value (unity odds ratio) was a sample size artefact because the significant ability of the markers to discriminate clinical statuses was shown in the multivariate (partial least squared) regression. Abbreviations: D28, day 28; LDH, lactate dehydrogenase; Pa/Fi, quotient between the peripheral oxygen saturation and inspiratory oxygen fraction; ROX, ROX index (the ratio of Pa/Fi to the respiratory rate); V4, visit 4; V5, visit 5; WHO, World Health Organization.

**Figure 3 pathogens-14-00533-f003:**
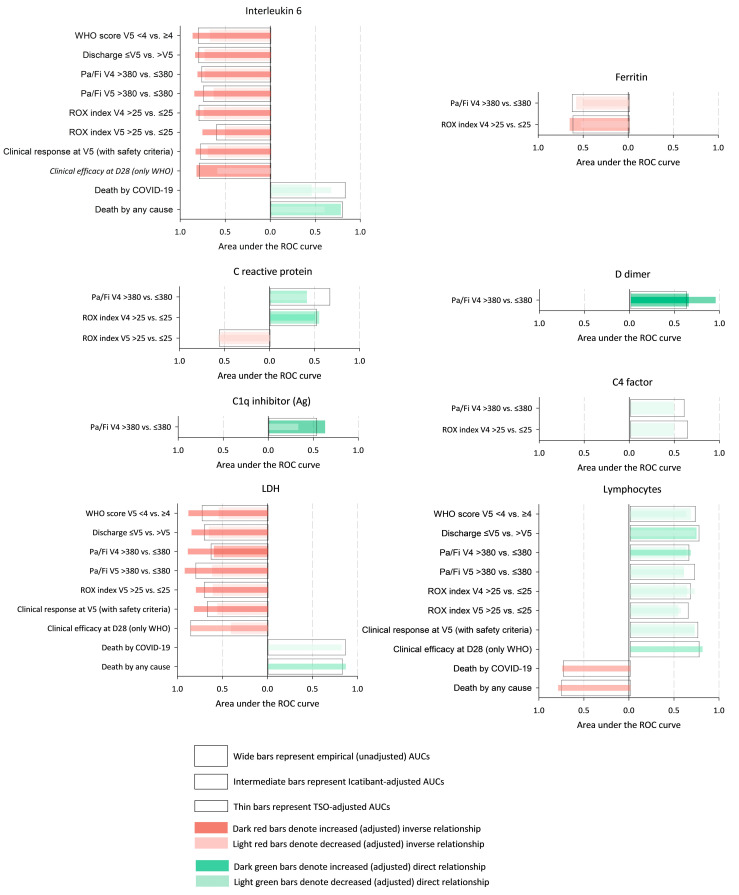
Representation of the empirical and adjusted discriminative ability of the markers for discerning clinical milestones for each of the 38 marker–milestone pairs analysed. The bars represent the areas under the empirical (clear bars) and adjusted (coloured bars) ROC curves of the discriminative ability of each marker to discern between the dichotomised clinical milestones for each of the 38 marker–milestone pairs identified in the partial least squares regression. Red bars denote inverse relationships (the greater the value of the marker, the lower the probability of the clinical event), whilst green bars denote direct relationships (the greater the value of the marker, the higher the probability of the clinical event). Abbreviations: Ag, antigenic; AUC, area under the curve; D28, day 28; LDH, lactate dehydrogenase; ROC, receiver operating characteristic; Pa/Fi, quotient between the peripheral oxygen saturation and the inspiratory oxygen fraction; ROX, ROX index (the ratio of Pa/Fi to respiratory rate); TSO, time since symptom onset; V4, visit 4; V5, visit 5; WHO, World Health Organization.

**Figure 4 pathogens-14-00533-f004:**
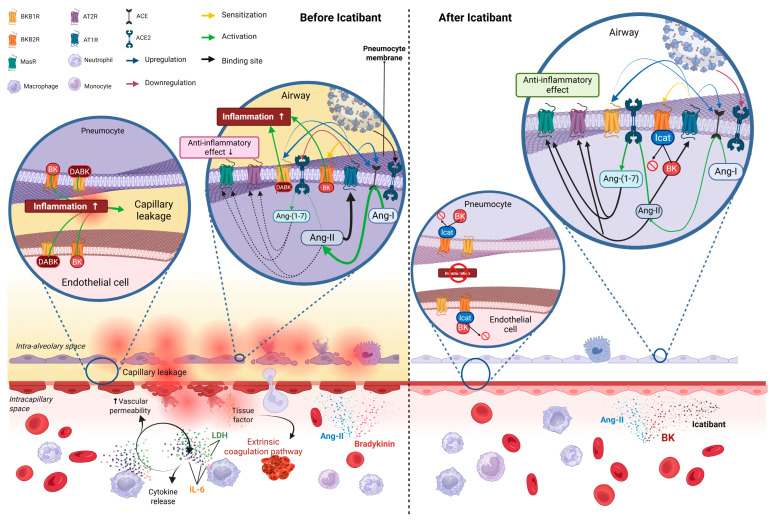
Summary of COVID-19 pathophysiology: lessons learned from this study after exposure to icatibant. Left panel: Since the BK/BKB2R pathway generates inflammation (cytokine release and white blood cell recruitment), vasodilation (with capillary leakage and oedema formation), ACE2 downregulation (thin dashed arrows), and ACE upregulation (continuous thick arrows), icatibant (right panel), a selective competitive BKB2R antagonist, would break one of the inputs of the vicious proinflammatory feedback loop created by the SARS-CoV-2 infection. The higher availability of ACE2 should increase the Ang-(1–7) levels by metabolising Ang-II, increasing the ACE2/MasR pathway anti-inflammatory signalling. Also, less circulating Ang-II should reduce the AT1R signalling, thus reducing its synergistic proinflammatory augmentation. Abbreviations: ACE, Angiotensin-Converting Enzyme; Ang, Angiotensin; AT1R, Angiotensin II Type 1 Receptor; AT2R, Angiotensin II Type 2 Receptor; BK, Bradykinin; BKB1R, Bradykinin B1 Receptor; BKB2R, Bradykinin B2 Receptor; DABK, des-Arg^9^-Bradykinin; Icat, Icatibant; IL-6, Interleukin 6; MasR, Mas Receptor.

**Table 1 pathogens-14-00533-t001:** Characteristics of the patients at the baseline.

	Icatibant Group (*n* = 37)	SoC (Control) Group (*n* = 36)
Median age (IQR)—years	49.0 (41.0–59.0)	56.5 (46.8–70.2)
Male sex—no. (%)	27 (73.0)	22 (61.1)
Median body mass index (IQR)—kg/m^2 1^	28.2 (25.7–36.3)	30.3 (26.3–33.0)
Median respiratory rate (IQR)—breaths/minute	21.0 (18.0–25.0)	20.0 (18.0–22.0)
$\mathrm{Median}\;{Pa}_{O_{2}}$ (IQR)—mmHg ^2^	71.0 (65.0–84.0)	71.0 (63.8–79.2)
$\mathrm{Median}\;{Fi}_{O_{2}}$ (IQR)—unitless	0.28 (0.21–0.32)	0.28 (0.21–0.33)
$\mathrm{Median}\;{Pa}_{O_{2}}$$/{Fi}_{O_{2}}$ (IQR)—unitless	261.0 (212.0–317.0)	263.0 (210.0–320.0)
${Sp}_{O_{2}}$ (IQR)—% ^3^	96.0 (95.0–97.0)	96.0 (95.0–97.0)
$\mathrm{Median}\;{Fi}_{O_{2}}$ (IQR)—unitless ^3^	0.30 (0.28–0.35)	0.31 (0.28–0.35)
$\mathrm{Median}\;{Sp}_{O_{2}}$$/{Fi}_{O_{2}}$ (IQR)—unitless ^3^	326.0 (274.0–346.0)	308.0 (275.0–343.0)

^1^ The body mass index is the weight in kilograms divided by the square of the height in metres. ^2^ The conversion factor to *Système International* units (kPa) is 0.133. ^3^ Measured under oxygen supplementation. Abbreviations: ${Fi}_{O_{2}}$, inspiratory oxygen fraction; IQR, interquartile range; kg, kilogram; m, metre; no., number; mmHg, millimetres of mercury; ${Pa}_{O_{2}}$, partial (arterial) oxygen pressure; SoC, standard of care; ${Sp}_{O_{2}}$, peripheral oxygen saturation.

## Data Availability

The data presented in this study are available on request from the corresponding author because this is a post hoc analysis of a clinical trial published elsewhere.

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
