# Peer review of "Pathophysiology of COVID-19: A Post Hoc Analysis of the ICAT-COVID Clinical Trial of the Bradykinin Antagonist Icatibant"

_pathogens, 2025, doi:10.3390/pathogens14060533_

Round 1
Reviewer 1 Report
Comments and Suggestions for Authors This paper explores the pathophysiological mechanisms behind severe COVID-19 by analyzing data from the ICAT-COVID trial, which tested the bradykinin B2 receptor antagonist icatibant. The authors identify IL-6, LDH, and lymphocyte count as key biomarkers and show how icatibant modulates their predictive value for clinical outcomes. I think addressing the following concerns can improve the paper: 1. The paper can benefit from a clearer explanation of how post-hoc analysis limitations might affect the reliability of conclusions, especially since causality cannot be firmly established. 2. The authors should discuss in more detail why some known biomarkers like D-dimer and ferritin underperformed in their analysis. 3. The graphical figures and ROC curve panels are dense and may overwhelm readers; summarizing the main findings in a simplified figure or table could help. 4. I found the discussion on timing of icatibant treatment very relevant—this point deserves more emphasis in the abstract and conclusions. 5. Some methodological details ( the multivariate model assumptions or ROC calculation choices) need clearer justification to boost transparency and reproducibility.Author Response
This paper explores the pathophysiological mechanisms behind severe COVID-19 by analyzing data from the ICAT·COVID trial, which tested the bradykinin B2 receptor antagonist icatibant. The authors identify IL-6, LDH and lymphocyte count as key biomarkers and show how icatibant modulates their predictive value for clinical outcomes. I think addressing the following concerns can improve the paper.
Comment 1: The paper can benefit from a clearer explanation of how post hoc analysis limitations might affect the reliability of conclusions, especially since causality cannot be firmly established.
Response 1: Thank you for pointing this out. In response to your comment, we have included some extra sentences at the beginning of the limitations paragraph to provide more details on the implications of the exploratory nature of this research (pages 7-8, lines 233-238 of the revised manuscript). We hope that the explanations included are satisfactory to you.
Comment 2: The authors should discuss in more detail why some known biomarkers like D-dimer and ferritin underperformed in their analysis.
Response 2: Following your suggestion, we have expanded a little further in the discussion on the potential reasons for the lack of discriminative ability of D-dimer, ferritin and, in general, acute-phase reactants (page 9, lines 285-286, 289-292, and 296-299 of the revised manuscript).
Comment 3: The graphical figures and ROC curve panels are dense and may overwhelm readers; summarizing the main findings in a simplified figure or table could help.
Response 3: We acknowledge that the ROC curve panels are dense; for that reason, we have placed them in the supplementary materials, just for the interested readers and for the sake of transparency and reproducibility. Precisely, to avoid overwhelming readers, we have tried to produce a concise and intuitive overview of all findings by using the butterfly plots included on Figure 3. Please, note that this figure, which was the result of thorough efforts to summarize the main findings into a single device, already provides information on both the directionality and modulation of the discriminative ability by icatibant treatment of all markers at a single glance, which is just what you suggested. Therefore, we think that no further action is needed following this comment. Nevertheless, if you find that the extensive ROC panels are excessive, even as supplementary material, we could remove them, since are not vital for drawing the conclusions.
Comment 4: I found the discussion on timing of icatibant treatment very relevant–this point deserves more emphasis in the abstract and conclusions.
Response 4: We appreciate the positive appraisal of the discussion about icatibant timing and the suggestion to acknowledge it in the abstract and conclusions. In consequence, we have updated both sections to mention that icatibant treatment should be given early during the acute hyperinflammatory stage of the disease, before the respiratory distress syndrome fully unfolds (page 1, lines 35-36 and page 10, lines 338-340 of the revised manuscript).
Comment 5: Some methodological details (the multivariate model assumptions of ROC calculation choices) need clearer justification to boost transparency and reproducibility.
Response 5: Empirical ROC curves do not require modeling assumptions since they can reduce to the mapping of the (empirical) survival functions of observed marker values between the negative (healthy, “non-diseased”) and positive (“diseased”) populations. However, modeling covariate effects on test results or ROC curves actually requires some parametric assumptions because it goes through preliminary modelling of marker results. In particular, we assumed that they admitted a Normal parametrization, which we deem is not very unreasonable in view of the results of the adjusted analyses (see the Figures S10 to S18 in the Supplementary Material). On the other hand, the binormal ROC model used for evaluating the effects over ROC curves themselves provides a robust parametrization of empirical ROC curves in the sense that it does not require that biomarker results follow a Normal distribution, because (empirical ROC curves) pertain to the relationships between the aforementioned survival functions rather than to the distributions of biomarker results themselves [1, 2]. In consequence, we think that the analyses done to produce the reported results do not suffer from major methodological flaws.
In response to your comment, we have included additional methodological details on the Supplementary Methods (Supplementary Materials, pages 12-13). We have preferred no to include them in the main document to avoid overwhelming clinicians with excessively technical details of statistics, which is in consonance with your third comment.
Reviewer 2 Report
Comments and Suggestions for Authors
Review Report
11.05.2025
I was invited to review the manuscript submitted to Pathogens-MDPI, entitled:
“Pathophysiology of COVID-19: A post-hoc analysis of the ICAT-COVID clinical trial of the bradykinin antagonist icatibant.”
The topic of the manuscript is highly relevant and contributes meaningfully to the understanding of the pathogenesis of SARS-CoV-2 infection. The article is well-structured and carefully prepared.
Comments and suggestions:
- In the methodology section, the inclusion criterion for patient selection was "requiring supplemental but not high-flow oxygen or mechanical ventilation." What was the oxygen saturation (SpOâ‚‚) threshold considered for initiating supplemental oxygen?
- What was the standard of care (SoC) for COVID-19 at your hospital during the study period? Please specify or cite the relevant protocol or guideline used.
- The small number of study participants and the limited statistical power of the analyses should be acknowledged as limitations of the study.
Additionally, the manuscript would benefit from a thorough English language revision by a native English editor
Author Response
I was invited to review the manuscript submitted to Pathogens-MDPI, entitled: “Pathophysiology of COVID-19: A post-hoc analysis of the ICAT-COVID clinical trial on the bradykinin antagonist icatibant.”
The topic of the manuscript is highly relevant and contributes meaningfully to the understanding of the pathogenesis of SARS-CoV-2 infection. The article is well structured and carefully prepared.
Comment 1: In the methodology section, the inclusion criterion for patient selection was “requiring supplemental but not high-flow oxygen or mechanical ventilation.” What was the oxygen saturation (SpO2) threshold considered for initiating supplemental oxygen?
Response 1: We performed arterial blood gas analysis in all patients; thus, we had direct measurements of partial (arterial) oxygen pressure available, which we used to calculate the / ratio. The threshold we used to initiate supplemental oxygen was based on such ratio (<380), which was also used as an inclusion criterion of the parent clinical trial. Using the default average atmospheric oxygen concentration of 21%, such ratio corresponds to an oxygen pressure of 80 mmHg. Correspondence analyses indicate that this pressure is approximately equivalent to 95% saturation by pulse oximetry.
Following your comment, we have provided some detail on respiratory physiology parameters included in the selection criteria (page 3, lines 112-113 of the revised manuscript).
Comment 2: What was the standard of care (SoC) for COVID-19 at your hospital during the study period? Please specify or cite the relevant protocol or guideline used.
Response 2: The standard of care included: a) respiratory support measures, b) fluid therapy, c) antipyretic treatment, d) postural therapy, e) low molecular weight heparin, f) dexamethasone (if certain criteria were met), g) remdesivir (if certain criteria were met), and h) tocilizumab (if certain criteria were met).
In response to your comment, we have included a short mention to this in the description of the selection criteria (page 3, lines 116-119 of the revised manuscript).
Comment 3: The small number of study participants and the limited statistical power of the analyses should be acknowledged as limitations of the study.
Response 3: We thank you for pointing this out, since it had not been referred as a limitation of this research, and in fact it is. In response, we have included it in the limitations paragraph (pages 7-8, lines 233-238 of the revised manuscript). Please, see also the response to the first comment by the other reviewer.
Comment 4: Additionally, the manuscript would benefit from a thorough English language revision by a native English author.
Response 4: We have had the manuscript reviewed again by a native English teacher with experience in copy-editing biomedical manuscripts (changes throughout the document).